# Improved Detection of Molecular Markers of Atherosclerotic Plaques Using Sub-Millimeter PET Imaging

**DOI:** 10.3390/molecules25081838

**Published:** 2020-04-16

**Authors:** Jessica Bridoux, Sara Neyt, Pieterjan Debie, Benedicte Descamps, Nick Devoogdt, Frederik Cleeren, Guy Bormans, Alexis Broisat, Vicky Caveliers, Catarina Xavier, Christian Vanhove, Sophie Hernot

**Affiliations:** 1Laboratory of In Vivo Cellular and Molecular Imaging (ICMI, BEFY-MIMA), Vrije Universiteit Brussel, Laarbeeklaan 103, 1090 Brussels, Belgium; jessica.bridoux@vub.be (J.B.); Pieterjan.debie@vub.be (P.D.); ndevoogd@vub.be (N.D.); vicky.caveliers@uzbrussel.be (V.C.); Catarina.xavier@vub.be (C.X.); 2Preclinical imaging, MOLECUBES NV, 9000 Ghent, Belgium; sara.neyt@molecubes.com; 3IBiTech-MEDISIP, Ghent University, 9000 Ghent, Belgium; Benedicte.descamps@ugent.be (B.D.); Christian.Vanhove@ugent.be (C.V.); 4Radiopharmaceutical Research, KU Leuven, 3000 Leuven, Belgium; frederik.cleeren@kuleuven.be (F.C.); guy.bormans@kuleuven.be (G.B.); 5Radiopharmaceutiques Biocliniques, INSERM 1039, Université de Grenoble, 38400 Grenoble, France; alexis.broisat@inserm.fr; 6Nuclear Medicine department, UZ Brussel, 1090 Brussels, Belgium

**Keywords:** vulnerable plaque, molecular imaging, PET imaging, nanobody, single-domain antibody, sub-millimetre resolution, AlF-radiolabelling

## Abstract

Since atherosclerotic plaques are small and sparse, their non-invasive detection via PET imaging requires both highly specific radiotracers as well as imaging systems with high sensitivity and resolution. This study aimed to assess the targeting and biodistribution of a novel fluorine-18 anti-VCAM-1 Nanobody (Nb), and to investigate whether sub-millimetre resolution PET imaging could improve detectability of plaques in mice. The anti-VCAM-1 Nb functionalised with the novel restrained complexing agent (RESCA) chelator was labelled with [^18^F]AlF with a high radiochemical yield (>75%) and radiochemical purity (>99%). Subsequently, [^18^F]AlF(RESCA)-cAbVCAM1-5 was injected in ApoE^−/−^ mice, or co-injected with excess of unlabelled Nb (control group). Mice were imaged sequentially using a cross-over design on two different commercially available PET/CT systems and finally sacrificed for ex vivo analysis. Both the PET/CT images and ex vivo data showed specific uptake of [^18^F]AlF(RESCA)-cAbVCAM1-5 in atherosclerotic lesions. Non-specific bone uptake was also noticeable, most probably due to in vivo defluorination. Image analysis yielded higher target-to-heart and target-to-brain ratios with the β-CUBE (MOLECUBES) PET scanner, demonstrating that preclinical detection of atherosclerotic lesions could be improved using the latest PET technology.

## 1. Introduction

Atherosclerosis is the progressive narrowing of arteries caused by the accumulation of lipids and fibrous elements in the artery walls. Although lesion growth will lead to progressive blood vessel occlusion, a large proportion of patients show no sign of disease until the sudden rupture of so-called vulnerable plaques. Rupture is usually associated with thrombosis, causing myocardial infarctions, strokes or peripheral vascular disease [1]. Together, those severe clinical complications claim over 15 million lives every year, making cardiovascular diseases the leading cause of death worldwide [2]. The joint ESC Guidelines suggested that early diagnosis of high-risk patients could be equally effective as preventing new cases, leading to potential cost-savings, and consequently encouraged research regarding risk assessment by non- or minimally invasive imaging techniques [3]. Techniques such as multi-detector CT [4,5], intravascular ultrasound [6,7], MRI [8,9], or Optical Coherence Tomography [10] are extensively being investigated for the assessment of morphological or structural characteristics of atherosclerotic lesions. Among the molecular imaging modalities, which can reveal specific biological aspects of atherosclerotic plaques, positron emission tomography/computed tomography (PET/CT) is one of the preferred techniques in clinic [11]. Although PET/CT is a sensitive and quantitative technique, most of the commercially available pre-clinical PET scanners do not meet the necessary sensitivity and spatial resolution to fully support clinical translation of new promising tracers [12]. Recently a novel PET scanner (β-CUBE, Molecubes, Ghent, Belgium) became commercially available that uses monolithic scintillation detectors to obtain sub-mm spatial resolution in combination with high sensitivity [13], which might improve plaque detection in mice.

In order to visualise the recruitment of inflammatory cells in atherosclerotic plaques with PET imaging, we used a Nanobody (Nb)-based tracer (cAbVCAM1-5) targeting the vascular cell adhesion molecule-1 (VCAM-1) [14,15]. Nbs are small antigen-binding fragments derived from heavy-chain-only antibodies and proved to have ideal characteristics for PET imaging [16,17]. Furthermore, the biological half-life of Nbs matches the half-life of fluorine-18 (^18^F) (109.8 min), the most commonly used positron-emitting isotope because of its favourable nuclear decay characteristics. ^18^F-labelling of heat-sensitive biomolecules is commonly performed via prosthetic groups. However, this time-consuming process often has low efficiency. Herein, we overcame the previous issues by functionalising the Nb with the novel restrained complexing agent (RESCA) developed by Cleeren et al. [18], allowing fast and simple ^18^F-labelling via the Al^18^F-method [19,20].

In this study, the cAbVCAM1-5 Nb, was labelled via Al^18^F(RESCA) chemistry, and evaluated as a tracer to image atherosclerosis plaques in Apolipoprotein E-deficient (ApoE^−/−^) mice. In addition, we investigated whether the imaging could be improved using the latest β-CUBE PET technology.

## 2. Results

### 2.1. Conjugation with RESCA and Radiolabelling of the Nb

The produced cAbVCAM1-5 Nb was randomly modified through conjugation of tetrafluorophenyl TFP-RESCA (Figure 1) on its lysines for subsequent Al^18^F-labelling. Electrospray ionisation and quadrupole time-of-flight mass spectrometry (ESI-Q-ToF-MS) analysis revealed successful conjugation of the cAbVCAM1-5 Nb with RESCA. For cAbVCAM1-5-(RESCA)n, a mass of 14,658 + *n* × 419.5 Da was expected. Measured mass was obtained for *n* = 1 (15,076 ± 2) Da, *n* = 2 (15,495 ± 2) Da, *n* = 3 (15,913 ± 2) Da and *n* = 4 (16,331 ± 2) Da (Appendix A).

Next, cAbVCAM1-5 randomly conjugated with RESCA was radiolabelled at room temperature (RT) with [^18^F]AlF with a 78 ± 2% radiochemical yield (RCY). Separation of Nb from free [^18^F]AlF was performed through a desalting PD10 column which was eluted in 500 μL fractions. The two fractions containing most of the activity were combined and filtered, allowing to obtain a radiochemical purity (RCP) of 99% (Figure 2) and an apparent molar activity of 24.5 ± 3.1 GBq/μmol. The radiolabelling and purification procedures were completed in less than an hour. [^18^F]AlF(RESCA)-cAbVCAM1-5 Nb remained stable with a RCP of 96% (Appendix A) over 3 h 30 min in injection buffer at RT, as well as in human serum at 37 °C over 1 h 30. At 2 h 30 min up to 6% defluorination was observed in human serum (Appendix A).

### 2.2. Imaging with the β-CUBE and LabPET8 Systems

In vivo PET imaging showed excretion of the tracer via the kidneys and bladder. The cohort injected with the [^18^F]AlF(RESCA)-cAbVCAM1-5 Nb showed substantial signal in bone structures (Figure 3A, upper row). This signal was also observed in the control group (Figure 3A, lower row), where the [^18^F]AlF(RESCA)-cAbVCAM1-5 Nb was co-injected with excess of unlabelled cAbVCAM1-5 Nb, indicating the non-specific character of the uptake.

Accumulation of [^18^F]AlF(RESCA)-cAbVCAM1-5 Nb in the aortic arch of ApoE^−/−^ mice was observed, which is the predominant site for atherosclerotic lesion formation in this model (Figure 3A, upper row). No signal was observed in the aortic arch of the control group (Figure 3A, lower row). 

When comparing the imaging data obtained with two distinct preclinical PET devices in a crossover study, better image quality was achieved with the β-CUBE than with the LabPET8 (Figure 3A). In vivo image contrast was evaluated by calculating target-to-brain (T/B) and target-to-heart (T/H) ratios. In both cases, significantly higher values were obtained with the β-CUBE than with the LabPET8 (T/B: 3.88 ± 0.88 vs. 2.57 ± 0.54, *p* < 0.05; T/H: 1.75 ± 0.30 vs. 1.40 ± 0.24, *p* < 0.05; respectively).

### 2.3. Ex Vivo Biodistribution and Atherosclerotic Plaque Targeting of [^18^F]AlF(RESCA)-cAbVCAM1-5

The biodistribution of [^18^F]AlF(RESCA)-cAbVCAM1-5 is summarised in Figure 4A and Appendix A. Uptake in various organs and tissues is expressed as injected activity per gram (%IA/g). Constitutively VCAM-1 expressing organs such as the spleen (1.01 ± 0.34 %IA/g), lymph nodes (0.55 ± 0.15 %IA/g) and thymus (0.32 ± 0.09 %IA/g) showed specific uptake. These values were significantly lower when an excess of unlabelled Nb was co-injected (respectively 0.34 ± 0.14 %IA/g, 0.33 ± 0.22 %IA/g and 0.22 ± 0.06 %IA/g). In corroboration with the imaging data, high bone uptake was observed, which could not be reduced by competition (1.13 ± 0.33 vs. 0.96 ± 0.33 for the control). Other organs and tissues, except the kidneys (14.00 ± 3.75 %ID/g), showed no uptake of the tracer. Analysis of the dissected aortas and gamma counting confirmed the specific lesion targeting with [^18^F]AlF(RESCA)-cAbVCAM1-5 Nb as seen on the PET/CT images. Uptake in the aortas of ApoE^−/−^ mice was 2.15 ± 0.06 times higher (*p* < 0.03) as compared to the control group (Figure 4B). This was further confirmed by autoradiography of the dissected aortas even though some background in the blocked group is visible due to non-specific binding (Figure 4C).

## 3. Discussion

In the present study, as a proof of concept, a preclinical PET scanner capable of achieving sub-mm spatial resolution was used to image VCAM-1 expression in atherosclerotic lesions of ApoE^−/−^ mice using an Al^18^F(RESCA)-labelled Nb.

The lead compound cAbVCAM1-5 has previously been reported to target both the murine and human VCAM-1 receptor with nanomolar affinity, and was originally validated as technetium-99m (^99m^Tc)-labelled tracer for SPECT imaging by Broisat et al. [14]. Although new generation clinical SPECT cameras are emerging [21], PET imaging remains the preferred imaging modality to quantitatively image molecular markers in the clinic. In addition, short-lived isotopes commonly used for PET imaging, such as fluorine-18 (^18^F) or gallium-68 (^68^Ga), match the short biological half-life of Nbs, allowing to decrease the radiation burden for the patients. To this end, the cAbVCAM1-5 Nb was radiolabelled with ^18^F via the N-succinimidyl 4-[^18^F]-fluorobenzoate ester ([^18^F]SFB) prosthetic group [22]. Despite excellent in vivo results such as lower kidney retention and lower specific organ uptake, the production of [^18^F]SFB remains a time-consuming multi-step procedure. In an attempt to facilitate its clinical translation, analogues of the cAbVCAM1-5 Nb labelled with radiometals via chelation were studied by Bala et al. [23] ^68^Ga, however, has some disadvantages such as a very short half-life of 68 min, limiting the number of patients per synthesis, and requires a ^68^Ge/^68^Ga generator which is not available at every hospital. More recently, a new bifunctional chelator, RESCA-tetrafluorophenyl ester (TFP-RESCA) has been developed by Cleeren et al. [18,19]. Contrarily to most fluorination methods, the RESCA chelator, allows fast radiolabelling of biomolecules with [^18^F]AlF at RT in aqueous solution.

We applied this strategy by coupling the RESCA chelator to the cAbVCAM1-5 Nb via its lysine residues, and an excellent radiolabelling yield with [^18^F]AlF was obtained (>75% RCY in less than 60 min) (in comparison, radiolabelling of Nbs using [^18^F]SFB have global yields ranging between 5 and 15% for a procedure time of 180 min) [24].

PET imaging of atherosclerotic lesions in the aortic arch of mice could be performed 2.5 h after intravenous (IV) administration of the radiolabelled Nb. Ex vivo analysis confirmed the imaging data, showing a significantly higher uptake of the tracer in excised aortas (0.42 ± 0.08 %IA/g) compared to the blocking condition (0.20 ± 0.02 %IA/g). This uptake, however, is lower than the previously reported uptake with [^18^F]FB-labelled cAbVCAM1-5 Nb (1.18 ± 0.36 %IA/g) [22].

The non-invasive detection of small atherosclerotic lesions could be improved using a preclinical PET scanner of the latest generation (such as the β-CUBE), yielding significantly higher plaque-to-brain and plaque-to-heart ratios. This confirms the importance of PET scanners with sub-mm spatial resolution and high sensitivity to evaluate novel tracers for atherosclerotic plaque detection and characterisation.

Uptake of the radiolabelled cAbVCAM1-5 Nb in constitutively VCAM-1 expressing organs such as the spleen and the thymus was expected since this was already observed in previous studies [14,22,23]. This is attributed to specific targeting because uptake could be prevented by an excess of the cold analogue. Contrarily, bone uptake was noticeably high for both experimental groups. Although some specific uptake due to the expression of VCAM-1 in the bone marrow could be expected [25], the signal was observed in both groups, indicating the non-specific character of the uptake. The PET/CT images accurately co-localised the signal with bone structures (e.g., skull, limb bones, and vertebral column and sternum) and is most likely a result of uptake of degradation products derived from the tracer. The nature of this degradation product could either be formation of an active metabolite, or most likely in vivo decomplexation of [^18^F]AlF, or defluorination, considering that radiometals and free fluorine tend to accumulate in the bone structures [26]. A hypothesis could be that after glomerular filtration and reabsorption in the proximal tubuli, the [^18^F]AlF-RESCA complex is degraded in the lysosomes where the Nb is internalised. This results in the release of [^18^F]AlF or [^18^F]F^−^ that returns into blood circulation and accumulates in the bone structures. As this instability appears to be lower in the case of other proteins [20,27], it would be interesting to further investigate the reasons behind the degradation in the case of Nbs.

In the context of atherosclerosis imaging, this bone uptake is particularly undesirable for two reasons. First of all, imaging must be performed at a time point that ensures the lowest achievable blood background signal, while keeping a significant signal in the VCAM-1-low-expressing targeted plaque. Even though Nbs are cleared very quickly from the blood, previous studies showed that imaging 2.5 to 3 h after tracer injection for the cAbVCAM1-5 Nb resulted in an optimal target-to-blood ratio [14] which unfortunately correlates with the formation of radio-metabolites observed during in vivo biodistribution studies. Secondly, [^18^F]NaF is already being investigated in the clinic to image active calcification in atherosclerosis [28]. Thus, if free [^18^F]F^−^ or [^18^F]AlF is entering the blood, it would become unclear which biological process is being imaged.

## 4. Material and Methods

All reagents and solvents were purchased from Sigma–Aldrich (Overijse, Belgium) or VWR (Oud-Heverlee, Belgium). The RESCA bifunctional derivative was synthetised as described in patent WO/2016/065435 [18]. All buffers used for derivatisation and labelling of the Nb were prepared in metal-free water (Fluka, Honeywell, Brussels, Belgium), chelexed (chelex, 100 sodium form (Sigma Aldrich, Overijse, Belgium), 2 g/L, overnight shaking at RT) and filtrated with a 0.2 μm PES membrane filter (VWR). ^18^F was produced on site using a cyclotron (Cyclone KIUBE, IBA, Ottignies-Louvain-la-Neuve, Belgium) by irradiation of H2^18^O with 18-MeV protons. Radioactivity was measured using an ionisation chamber-based dose calibrator (Veenstra Instruments, Joure, The Netherlands).

### 4.1. Nb Production

The cAbVCAM1-5 Nb was generated by immunisation of a dromedary in the context of a previous study, was produced in E. coli as a *C*-terminal His-tagged (six histidine residues) protein and was purified by immobilised metal affinity chromatography and size exclusion chromatography according to standard procedures as described elsewhere [14].

### 4.2. Random-Conjugation of the RESCA Chelator

The cAbVCAM1-5 Nb was modified by incubation with a 12-fold molar excess of TFP-RESCA (135 mM stock in DMSO) in 0.05 M sodium carbonate buffer (120 μM of Nb, pH 8.5–8.7) for 2 h at RT. The modified Nb was subsequently purified by SEC on a Superdex Peptide 10/300 GL column (GE Healthcare, Belgium) and eluted in 0.1 M NH_4_OAc pH 7 on a medium-pressure chromatography system (Bio-Rad NGC, Belgium). The average number of chelators per Nb was determined by ESI-Q-ToF-MS (GIGA Proteomics, Liège, Belgium).

### 4.3. [^18^F]NaF Production

The 2-mL solution of [^18^F]F^−^ in enriched water was manually applied to a preconditioned Sep-Pak Light QMA cartridge (WAT023525, Waters, Zellik, Belgium). The QMA was rinsed with 3 mL of water and [^18^F]NaF was eluted with 300 μL of 0.9% NaCl solution (typically 5 GBq is eluted). The eluate was diluted with water to obtain a 10 MBq/μL solution.

### 4.4. Al^18^F-Labelling of the Nb-RESCA

100 µL (1 GBq) of [^18^F]NaF solution was added to 10 μL (20 nmol) of AlCl_3_ solution (2 mM AlCl_3_ trace-metal in 0.1 M NaOAc buffer pH 4.1) and left at RT for 5 min. The RESCA-conjugated cAbVCAM1-5 Nb (43 nmol, stock solution of 86.2 μM) in 0.1 M NH_4_OAc pH 4.5 was added to the reactor containing the previously prepared [^18^F]AlF solution and left at RT for 12 min.

### 4.5. Purification and Quality Control

The radiolabelled Nb was purified using a disposable PD-10 column (GE Healthcare, Machelen, Belgium) pre-conditioned with injection buffer (0.9% NaCl + 5 mg/mL vit. C, pH 6), and passed through a 0.22 µm PVDF membrane filter (Millex GV, Millipore, Darmstadt, Germany) before further use. RCP was assessed through instant thin layer chromatography (iTLC) on silica gel impregnated glass fiber sheets (Agilent Technologies, Machelen, Belgium) with 0.9% NaCl as mobile phase. iTLCs were analysed with a radio-TLC detector (RITA, Elysia Raytest, Angleur Belgium). RCP before in vivo injection was >99%. DC-RCY was calculated based on the activity obtained after PD-10 to the amount of starting activity used for Al18F-production, decay-corrected for the same time point.

### 4.6. In Vitro Stability Studies

At different time points, aliquots of filtered [^18^F]AlF(RESCA)-cAbVCAM1-5 Nb were analysed for stability in injection buffer or in serum. For the latter, 400 μL of filtered [^18^F]AlF(RESCA)-cAbVCAM1-5 Nb in injection buffer was added to 500 μL of human serum (Innovative research, Peary Court, FL, USA) and incubated at 37 °C. Analyses were performed via SEC on a HPLC system (Hitachi, Zaventem, Belgium) equipped with a radio-detector (GABI, Elysia Raytest, Angleur, Belgium) and on a Superdex 75 10/300 GL column (GE Healthcare, Machelen, Belgium) equilibrated with Phosphate Buffer Saline (PBS) ([^18^F]AlF(RESCA)-cAbVCAM1-5 Rt = 28.6 min, free [^18^F]AlF and [^18^F]F-Rt = 35.4 min).

### 4.7. Animal Model and Experimental Setup

All animal experiments were performed in accordance to the European guidelines for animal experimentation under the license LA1230272 and approved by the local Ethical Committee of the Vrije Universiteit Brussel (14-272-7). ApoE^−/−^ mice were obtained from Charles River (L’Abresle, France). ApoE^−/−^ mice were fed a high-fat Western diet with 1.25% cholesterol (D12108C, Research Diets, New Brunswick, NJ, USA) for 25–30 weeks to induce atherosclerotic lesions. VCAM-1 expression was assessed for this model in a previous study [14]. ApoE^−/−^ mice (*N* = 6) were injected IV with (14.52 ± 8.98) MBq (12 μg) of [^18^F]AlF(RESCA)-cAbVCAM1-5 Nb. The control group (*N* = 6) was injected with an excess of unlabelled cAbVCAM1-5 Nb (1.1 mg, 90-fold excess) followed by the injection of (14.34 ± 8.28) MBq (12 μg) of [^18^F]AlF-RESCA-cAbVCAM1-5 Nb 15 min after the first injection.

### 4.8. In Vivo PET/CT Imaging and Image Processing

Two hours and 30 min after tracer injection (as determined to be the optimal time point for ideal T/B ratio in a previous study) [14], mice (*N* = 6/group) were imaged sequentially on two different PET/CT systems using a cross-over design: a β-CUBE (Molecules, Ghent, Belgium) providing sub-mm (0.83 mm) spatial resolution and a LabPET8 (TriFoil Imaging, Chatsworth, CA, USA) with 1.2 mm spatial resolution. Both PET scans were acquired in list-mode with a total acquisition time of 30 min on the β-CUBE and 30 min on the LabPET8. PET data were reconstructed iteratively (OSEM for β-CUBE; MLEM for LabPET8) with a total of 50 iterations into a voxel size of 0.4 and 0.5 mm for the β-CUBE and LabPET8, respectively. Each PET scan was followed by a CT scan acquired for co-registration purposes using the CT-device of the same manufacturer (X-CUBE for β-CUBE; XO-CT for LabPET8). Volumes-of-interest were drawn at the level of the aortic arch, brain and heart, and T/B and T/H ratios were calculated.

### 4.9. Ex Vivo Analysis

Following in vivo PET/CT imaging, mice were euthanised to collect organs and tissues of interest. Aortas from the aortic root to the iliac bifurcation were collected as well. All samples were weighed and counted for radioactivity against a standard of known activity. Uptake was expressed as percentage of injected activity per gram (%IA/g), corrected for decay and extra-venous injection. Ex vivo autoradiography images were obtained after overnight exposure of the aortas to a dedicated phosphorscreen (Typhoon FLA 7000, GE Healthcare). Images were analyzed with ImageQuant.

### 4.10. Data Analysis and Statistics

Data are expressed as mean ± standard deviation. Comparisons between groups were performed using unpaired Student’s t-test; for comparisons between scanners a paired Student’s t-test was used. A *p*-value ≤ 0.05 was considered significant. Statistical analysis was performed using Prism 5 (Graph Pad Software) or SPSS Statistics software (version 24.0.0, IBM Company, Brussels, Belgium).

## 5. Conclusions

The cAbVCAM1-5 Nb could be easily radiolabelled with [^18^F]AlF through chelation with RESCA. The potential of ^18^F-labelled cAbVCAM1-5 Nb to target the atherosclerotic lesions and to provide good target-to-background ratios was demonstrated using the new β-CUBE imaging system. However, in vivo degradation leads to bone uptake of fluorine-18, which could interfere with the interpretation of imaging results. The excellent and uniform spatial resolution of the β-CUBE resulted in improved image quality and allowed better quantification as compared with an imaging system with lower resolution.

## Figures and Tables

**Figure 1 molecules-25-01838-f001:**
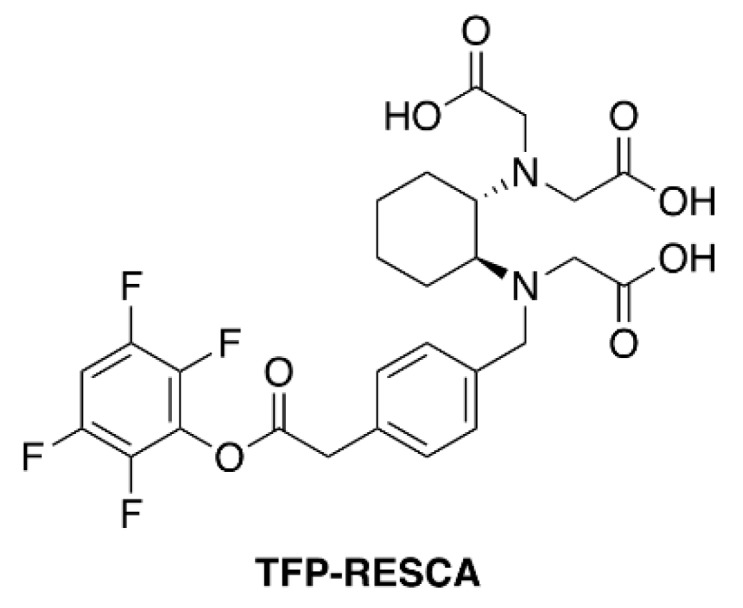
Structure of tetrafluorophenyl restrained complexing agent (TFP-RESCA).

**Figure 2 molecules-25-01838-f002:**
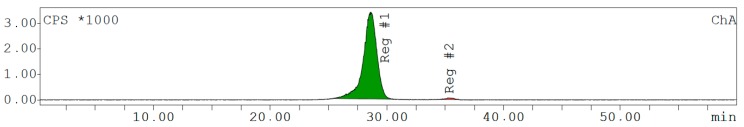
Size Exclusion Chromatography (SEC) profile of [^18^F]AlF(RESCA)-cAbVCAM1-5 Nb before injection. Retention time (Rt) of [^18^F]AlF(RESCA)-cAbVCAM1-5 = 28.7 min (99%), free [^18^F]AlF and [^18^F]F-Rt = 35.3 min (1%).

**Figure 3 molecules-25-01838-f003:**
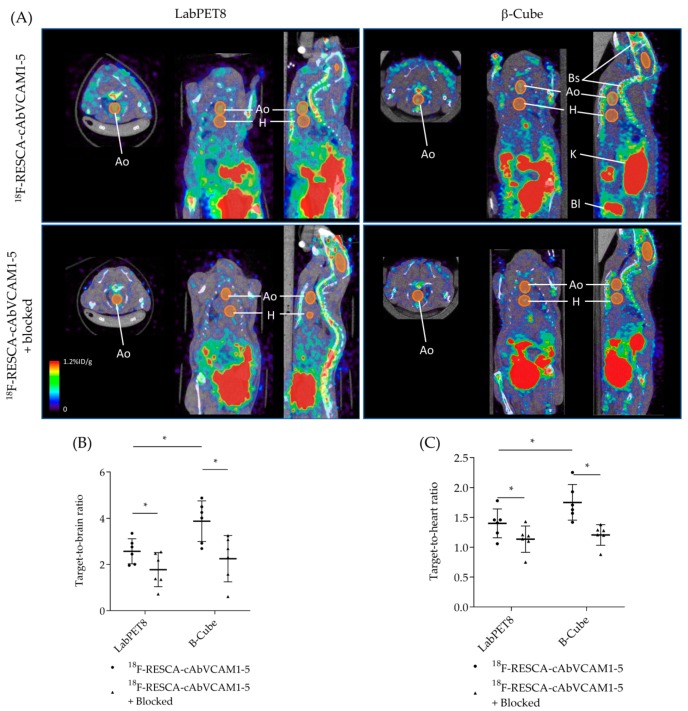
(**A**) Representative PET/CT images of the same mouse obtained with the LabPET8 (left) or β-CUBE (right) imaging system, demonstrating specific targeting of atherosclerotic lesions in the aortic arch (Ao) of ApoE^−/−^ mice injected with [^18^F]AlF(RESCA)-cAbVCAM1-5 Nb (upper row), while no uptake is seen at the level of the aortic arch of ApoE^−/−^ mice co-injected with a 90-fold excess of unlabelled cAbVCAM1-5 Nb (blocking condition as control, unlabelled excess injected 15 min before injection of radiolabelled Nb) (lower row). Kidneys (K), bladder (Bl) and bone structures (Bs) are also visible on the images. Target-to-brain (T/B) (**B**) and target-to-heart (T/H) (**C**) ratios were calculated to compare the image quality between two commercially available preclinical PET scanners (β-CUBE and LabET8). The number of asterisks in the figures indicates the statistical significance (* *P* < 0.05).

**Figure 4 molecules-25-01838-f004:**
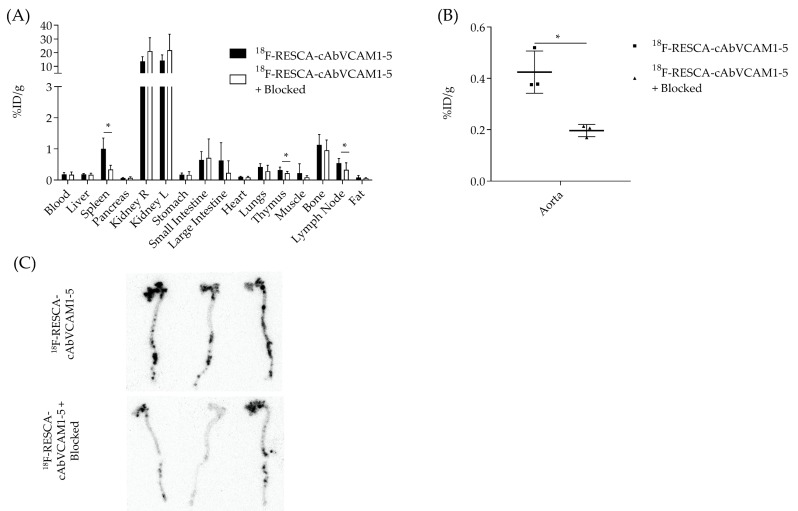
(**A**) Ex vivo biodistribution profile of [^18^F]AlF(RESCA)-cAbVCAM1-5 Nb in ApoE^−/−^ mice and ApoE^−/−^ mice co-injected with a 90-fold excess of unlabelled Nb (blocking condition). (**B**) Ex vivo analysis of excised atherosclerotic aorta, showing significantly higher uptake (2.15 ± 0.06 times; *p* < 0.03) of the [^18^F]AlF(RESCA)-cAbVCAM1-5 Nb compared to the control group (blocking condition). (**C**) Confirmation of the uptake by ex vivo autoradiography. The number of asterisks in the figures indicates the statistical significance (* *P* < 0.05).

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
