# Peer review of "Improved Detection of Molecular Markers of Atherosclerotic Plaques Using Sub-Millimeter PET Imaging"

_molecules, 2020, doi:10.3390/molecules25081838_

Round 1

Reviewer 1 Report

This manuscript is very interesting, and describes an original and promising way to label nanobodies. It moreover shows how radiopharmaceutical sciences benefit from chemistry (radiolabelling) and physics (detectors) progresses. The results are however a little bit disappointing, because of the moderate instability in vivo. This instability with innovative RESCA chelator was already known, for it was described in Cleeren’s paper with another nanobody (Cleeren F et al. Theranostics, doi: 10.7150/thno.20094). Interestingly, this instability appeared to be somehow lower with an affibody and with human serum albumin and has not been described with [18F]AlF-IL2 (van der Veen EL et al. Journal of Nuclear Medicine. https://doi.org/10.2967/jnumed.119.238782). It should thus be related with nanobodies’ behaviour in vivo. This might be interesting to investigate more deeply.

Beside this little drawback, I have several other comments:

  • A picture of RESCA chelator would have been fine.
  • In the introduction, line 52, the sentence is not clear and should reformulated.
  • In the results, line 103, β is missing.
  • In the discussion, line 166, I would add “such as” β-CUBE. The latter is not the only one latest generation preclinical PET scanner. Other very good ones are also available on the market…
  • In the materials and methods, line 202, remove “and [18F]AlF labelling”.
  • Line 203, remove “Nb production” as it is redundant with the above line.
  • In the 4.1 part, a brief description of the Nb production would be desirable.
  • In the 4.8 part, the authors should briefly explain why they chose to image at 2.5 h.
  • In the conflicts of interest, it should be mentioned that Sara Neyt is an employee of Molecubes NV

Author Response

Thank you very much for taking the time to review the paper and for your relevant remarks, here are the answers to the comments:

  • A picture of RESCA chelator would have been fine.

The picture has been added to the paper.

  • In the introduction, line 52, the sentence is not clear and should reformulated.

The sentence has been simplified.

  • In the results, line 103, β is missing.

I did not see the missing letter but it looks like it was added on line 103

  • In the discussion, line 166, I would add “such as” β-CUBE. The latter is not the only one latest generation preclinical PET scanner. Other very good ones are also available on the market…

Thank you for the remark, "such as" has been added  

  • In the materials and methods, line 202, remove “and [18F]AlF labelling”

It was removed

  • Line 203, remove “Nb production” as it is redundant with the above line.

It was also removed.

  • In the 4.1 part, a brief description of the Nb production would be desirable.

A few elements have been added to the sentence.

  • In the 4.8 part, the authors should briefly explain why they chose to image at 2.5 h

A small sentence mentioning the previous study has been added in brackets

  • In the conflicts of interest, it should be mentioned that Sara Neyt is an employee of Molecubes NV

This has been added

Reviewer 2 Report

Using a RESCA chelator to bind F18 and an anti-VCAM Nb the authors demonstrated using an in vivo model PET imaging on 2 platforms, BioD, and autoradiography the imaging of atherosclerotic plaques. Nicely done, good use of blocking antibody control and appropriate nuclide, Some minor comments.

Most importantly it would be nice to show in ex vivo analysis the presence of the plaques and their vcam-1 expression.
mention blocking condition and time post-tracer injection and remove conclusion statements in the figure legends.
Change bar charts to dots
Cite patent
on line 190 the citation seems to be incorrectly formatted.
Discuss why so high uptake in the brain.

Author Response

Thank you very much for taking the time review the paper and for your relevant remarks, here are the answers to the comments:

  • Most importantly it would be nice to show in ex vivo analysis the presence of the plaques and their vcam-1 expression.

We have demonstrated in different previous studies, which are referred in the paper, that the plaques in this model express VCAM-1 (Broisat et al. 2012). A short sentence have been added line 279 in the materials and methods section. 

Unfortunately this is not feasible anymore for that study since we do not have the model available at the moment and would be a time consuming experiment (25 - 30 weeks).

  • mention blocking condition and time post-tracer injection and remove conclusion statements in the figure legends.

Blocking condition and injection delay between non-radiolabelled and radiolabelled Nb have been added to the description of figure 3 (previously figure 2). The conclusion statement of figure 3 has been removed.

Blocking conditions have also been added in figure 4 (previously figure 3).

  • Change bar charts to dots

The figures showing uptake in the aortas, as well as the target to blood and brain ratio have been changed to dots charts.

  • Cite patent

The patent for RESCA is citation 18 and is referred in the materials and methods section.
We have added in the section conflict of interest the patent on the VCAM Nb.

  • on line 190 the citation seems to be incorrectly formatted.

It was corrected

  • Discuss why so high uptake in the brain.

We do not fully understand the comment of the reviewer about high brain uptake. According to us, there is not uptake in the brain tissue itself, however as we explained in the text, there is significant uptake in the bones including the skull. We assume that is what the reviewers refers to.
In the discussion text we had included a part to discuss skull and bone uptake, which was due to in vivo defluorination. The skull is mentioned in the discussion line 203 (in the corrected version).